# Dynamic Variability of Wind Erosion Climatic Erosivity and Their Relationships with Large-Scale Atmospheric Circulation in Xinjiang, China

**Yaqin Wang** [1,2], **Haimei Yang** [1,2], **Wenbo Fan** [2,*], **Changlu Qiao** [1,2] **and Kai Sun** [1,2]

1    College of Water and Architecture Engineering, Shihezi University, Shihezi 832000, China;
     wangyqqq0529@163.com (Y.W.); sjxy@shz.edu.cn (H.Y.); qiaochanglu@163.com (C.Q.);
     sk528713423@163.com (K.S.)
2    Key Laboratory of Modern Water-Saving Irrigation of Xinjiang Production & Construction Group,
     Shihezi University, Shihezi 832000, China
*    Correspondence: fwb205@163.com

**Abstract:** Xinjiang has a serious wind erosion problem due to its fragile ecological condition and sensitivity to climate change. Wind erosion climatic erosivity is a measure of climatic factors influencing wind erosion; evaluating its spatiotemporal variations and relationship with the large-scale circulation pattern can contribute to the understanding of the climate change effect on wind erosion risk. Thus, this study quantified the wind erosion climatic erosivity and examined the connections between climatic erosivity and climate indices using trend analysis, geo-statistical analysis, and cross-wavelet analysis based on the observed daily meteorological data from 64 weather stations in Xinjiang, China during 1969–2019 (50 years). The results indicated that the climatic erosivity showed a significant downward trend at seasonal and annual scales over the past 50 years. Strong seasonality in the C-factor was found, with its highest values in the spring and summer and its lowest values in the winter. The average climatic erosivity was weaker during El Niño events than during La Niña events. The impact of El Niño events on climatic erosivity in Xinjiang continued from the beginning of the event to two months after the end of the events. The La Niña events had a lag effect on the climatic erosivity in Xinjiang, with a lag period of 4 months. From a statistical perspective, the El Niño-Southern Oscillation (ENSO), North Atlantic Oscillation (NAO), and Arctic Oscillation (AO) indices showed relationships to the climatic erosivity in Xinjiang in terms of their correlation and periodicity. The relationships between the climatic erosivity and ENSO were not clearly positive or negative, with many correlations advanced or delayed in phase. The NAO and AO indices showed a consistent in-phase relationship with climatic erosivity on significant bands, whereas the profound mechanisms involved in this require further study. The results of this study provide a preliminary perspective on the effect of large-scale atmospheric circulation on wind erosion risk in arid and semi-arid regions.

**Keywords:** wind erosion climatic erosivity; cross-wavelet analysis; climate indices; Xinjiang

## 1. Introduction

Wind erosion is one of the most serious environmental issues in many arid and semi-arid regions of the world, which is the main cause of land degradation and desertification [1,2]. In China, about one-third of the territory, which is distributed in the arid region, is suffering from serious land desertification [3]. Wind erosion is a complex process that is affected by a large number of factors, including climate conditions, soil properties, land surface characteristics, and land-use practice [4–8]. Among these factors, climatic conditions are considered to be one of the most important driving forces in arid areas [9–11].

The effect of climatic conditions on wind erosion is not only reflected in the effect of wind, but the result of the combined effect of wind speed, precipitation, and tempera-

ture [12]. Chepil et al. believed that it is the climatic conditions that determine the annual level of soil wind erosion [13], and they proposed an index of wind erosion climatic factors (C-factor) that can represent and reflect the comprehensive effect of climatic conditions on wind erosion to estimate the amount of soil wind erosion under different climatic conditions [14,15]. However, the parameter setting and coefficient determination of the formula have strong regional limitations [16,17]. Then, the Food and Agriculture Organization of the United Nations (FAO) and Skidmore18 revised the model, respectively, introducing the ETP and probability density function of wind speed as parameters of the wind erosion climatic erosivity calculation model [18], which made up for the lack of theoretical basis in Chepil' s formula and reduced the calculation error in arid and semi-arid regions. However, Skidmore's formula involves many indicators, and it is difficult to collect calculation data, so it is not easy to popularize and apply [19]. The FAO version can be directly calculated using traditional meteorological data and meets well with the accuracy of the wind erosion equation. At present, the model is widely used in the assessment of wind erosion climatic conditions and response mechanism analysis in arid and semi-arid regions [20,21].

Changes in wind erosion climatic erosivity (C-factor value) are influenced by global climate change. Large-scale circulation patterns (El Niño-Southern Oscillation (ENSO), North Atlantic Oscillation (NAO), and Arctic Oscillation (AO)) have been the most prominent climate signals at the inter-annual scale during recent years and have had far-reaching consequences on global climate change [22,23]. ENSO is a periodic deviation in the expected sea surface temperature (SST) in the equatorial Pacific. The temperatures higher or lower than the normal ocean temperature can influence the weather patterns around the world by affecting the high–low pressure system, wind, and precipitation [24]. The occurrence of ENSO has a pronounced effect on most regions of China, especially on northern China, northeast China, southern China, Inner Mongolia, and Xinjiang, where the correlation has a good level of significance [25]. Scholars have found that ENSO events affect the hot and cold variability and dry–wet change in northwestern provinces, and the intensity of the variability is strongest in Xinjiang [26], where the variability in precipitation and temperature increases in the ENSO event years. In El Niño events, the humidifying effect in Xinjiang is obvious, and in La Niña events, there is a trend of drought in Xinjiang [27,28]. Some studies have noted that the annual NAO further affected the drought and flood events in Xinjiang by affecting the temperature and precipitation [29,30]. Certainly, the AO plays an important role in the variation in snow cover days and also had a significant correlation with snowfall and daily temperature extremes on an inter-decadal scale [31,32]. The impact of the above three climate indices for climate change is remarkable, but the impact of wind erosion climatic erosivity has rarely been analyzed.

Xinjiang is the main distribution area of the wind erosion landform in China, with drought and less rainfall, severe soil erosion, and a fragile ecological environment. Especially due to the increased climatic variability caused by the anomaly of large-scale atmospheric circulation, coupled with the strong impact of tropical cyclones, the ecological environment is extremely fragile. The study of the influence of ENSO remote correlation on wind erosion climatic erosivity in Xinjiang provides a theoretical basis for the comprehensive management of soil erosion and its prevention and control, which is of great significance for the monitoring, assessment, forecasting, and management of soil erosion. The results of this study will provide a preliminary perspective on the effect of large-scale atmospheric circulation on wind erosion risk in arid and semi-arid regions. Therefore, the objectives of this study are to (1) explore the characteristics of annual and seasonal wind erosion in Xinjiang under the background of climate change; and to (2) assess the relationship between wind erosion climatic erosivity and atmospheric circulation in Xinjiang.

## 2. Materials and Methods

### 2.1. Study Area

Xinjiang, located between $73°40'$–$96°23'$ E and $34°22'$–$49°10'$ N, has an area of $1.66 \times 10^6$ km$^2$, in which the desert area accounts for 24% of the total area [33]. Topographic

and geomorphological conditions in the region are complicated, forming an interlocking landform pattern of mountains, oases, and basins. The area is under a temperate continental arid and semi-arid climate, featured by a wide range of temperatures, strong wind, low and uneven distribution precipitation, and low humidity. Owing to the influence of the westerly circulation and the dry and cold airflow of the Arctic Ocean, more precipitation falls in Northern Xinjiang than in Southern Xinjiang, and more in West Xinjiang than in East Xinjiang. Coupled with the scarcity of natural vegetation and low coverage, the region's ecosystem is fragile [34]. As a result, the region is prone to catastrophic weather such as high winds, and environmental problems such as soil wind erosion and desertification are particularly prominent. Extreme wind erosion and frequent sandstorms wreak ecological havoc and cause environmental degradation in Xinjiang. Figure 1 shows the distribution of the meteorological stations.

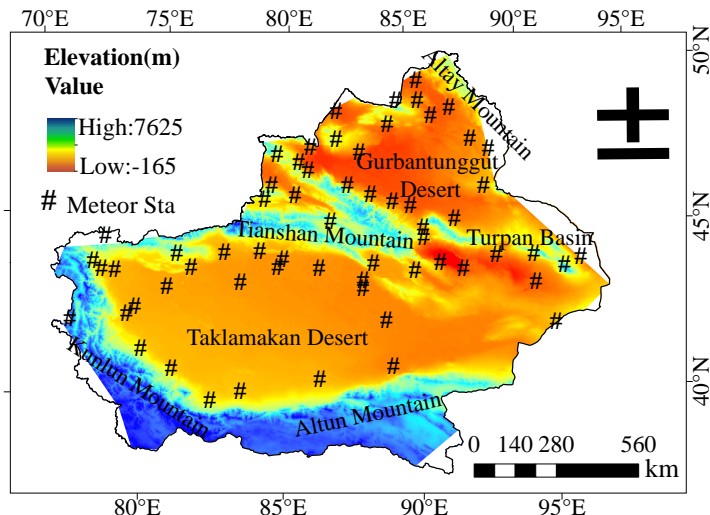

**Figure 1.** Location of the study area and distribution of the meteorological stations.

## 2.2. Data

Daily weather data including precipitation, maximum and minimum temperature, wind speed, and relative humidity were collected from the National Meteorological Information Centre of China (http://cdc.cma.gov.cn, accessed on 25 March 2021). After accounting for missing data and comparing the length of the recorded period, we then selected 64 stations for the period from 1969 to 2019 (50 years). The selected stations were relatively evenly distributed, which enabled the regional climatic change in Xinjiang to be reflected. For most of the stations (*n* = 45), time series of 51 years was available. The remaining stations had a series of at least 49 years. In order to ensure the integrity of the data and the accuracy of the calculation, the missing data were completed using the interpolation methods, including: (1) if only one day had missing data, the missing data would be replaced by the average value of its two nearest stations; (2) if two consecutive or more days had missing data, the missing data would be processed by simple linear correlation between its nearest stations. Data quality control was performed by the National Meteorological Information Center of China Meteorological Administration.

ENSO data were provided by the U.S. Atmospheric Administration (NOAA) Climate Prediction Center (CPC) (http://www.esrl.noaa.gov/, accessed on 11 August 2021). According to the method for the identification of El Niño/La Niña events [35], if the 3 month sliding average of SST anomalies in the Niño3.4 region reaches or exceeds 0.5 °C and lasts for at least 5 months, it is considered an El Niño event. If the 3 month sliding average falls below −0.5 °C and persists for at least 5 months, it is considered a La Niña event. The NAO and AO values between 1969 and 2019 were collected from the National Climate Center of China (https://cmdp.ncc-cma.net/cn/index.htm, accessed on 11 September 2021).

### 2.3. Computation of Wind Erosion Climatic Erosivity

The present study used the equation proposed by FAO to compute the wind erosion climatic factor [18]. The method is applicable to the estimation of wind erosion climatic erosivity in arid and semiarid areas, which is defined as follows:

$$C = \frac{1}{100} \sum_{i=1}^{12} \overline{u}^3 \left( \frac{ETP_i - p_i}{ETP_i} \right) d_i \qquad (1)$$

where $C$ is the wind erosion climatic factor; $\overline{u}$ is the monthly average wind speed at 2 m above the ground; $ETP_i$ is the potential evapotranspiration (mm) in month $i$; $p_i$ is the precipitation (mm) in month $i$; and $d_i$ is the number of days in month $i$. The potential evapotranspiration $ETP_i$ (mm) is calculated as described by reference [36]:

$$ETP_i = 0.19(20 + T_i)^2(1 - r_i) \qquad (2)$$

where $T_i$ is the monthly average air temperature (°C) and $r_i$ is the monthly average relative humidity (%).

### 2.4. Calculation of Climate Inclination Rate

The climate inclination rate method is used to analyze the inter-annual variation trend of wind erosion climatic erosivity. A climate variable with sample size $n$ is represented by $x_i$, and the time corresponding to $x_i$ is represented by $t_i$. The universal linear regression between $x_i$ and $t_i$ can be written as:

$$x_i = a + bt_i \ (i = 1, 2, \cdots, n) \qquad (3)$$

where $a$ is a regression constant, $b$ is a regression coefficient, $n$ is the sample size of a meteorological element, and $a$ and $b$ can be estimated by the least squares method [37]. According to the observed data $x_i$ and the corresponding time $t_i$, the least squares estimation of the regression the constant $b$ is:

$$b = \frac{\sum_{i=1}^{n} x_i t_i - \frac{1}{n}(\sum_{i=1}^{n} x_i)(\sum_{i=1}^{n} t_i)}{\sum_{i=1}^{n} t_i^2 - \frac{1}{n}(\sum_{i=1}^{n} t_i)} \qquad (4)$$

where $b \times 10$ is the climate inclination rate; when $b > 0$, the meteorological element sequence increases with time; otherwise, it decreases. The size of the $b$ value reflects the degree of the tendency to rise or fall. The correlation coefficient between time $t_i$ and variable $x_i$ can be written as:

$$r = \sqrt{\frac{\sum_{i=1}^{n} t_i^2 - \frac{1}{n}(\sum_{i=1}^{n} t_i)^2}{\sum_{i=1}^{n} x_i^2 - \frac{1}{n}(\sum_{i=1}^{n} x_i)^2}} \qquad (5)$$

The significance test of correlation coefficient is used to determine whether the degree of change trend is significant. Determining the level of significance, $\alpha$, if $|r| > r\alpha$, it indicates that the change trend of climate elements with time is significant; otherwise, it indicates that the change trend is not significant.

### 2.5. Statistical Methods

In this study, the spatial distribution map of the wind erosion climatic erosivity and its inclination rate for Xinjiang was completed by using the ordinary Kriging interpolation provided by the Geo-statistical Analyst Tool in ArcGIS 10.8 (It was developed by American Institute of Environmental Systems, Redlands, CA, USA). The limitation of Kriging interpolation is that its interpolation accuracy varies with the amount of sample data.

The cross-wavelet transform (CWT) can better reflect the phase structure and detailed characteristics in the time and frequency domain between two time series [38], and effectively analyze the correlation between them, but it can only reveal the phase relationship

between the two sequences in a high-energy region, while wavelet coherence (WTC) can make up for the lack of phase relationship analysis of the cross-wavelet in a low-energy region [39]. The cross-wavelet transform combined with wavelet coherence analysis can analyze the multi-time-scale correlation of two sequences in the time and frequency domain. Therefore, in this study, the cross-wavelet transform and coherent wavelet spectrum were used to analyze the multi-scale correlation between the wind erosion climatic erosivity and climatic index (ENSO, NAO, and AO).

## 3. Results

### 3.1. Annual and Seasonal Variation of Climatic Erosivity

The C-factor value is a measure of the climatic conditions most conducive to wind erosion. Its seasonal and annual change trend can reflect the change in soil wind erosion under the climatic characteristics for different periods of the year (Figure 2). Regardless of the seasonal or annual scale, climatic erosivity in Xinjiang exhibited a fluctuating downward trend. As far as the C-factor value of annual, spring, summer, and autumn, the whole region showed a significant downward trend ($p < 0.05$), and the decline rates were 0.33/a (annual), 0.19/a (spring), 0.11/a (summer), and 0.08/a (autumn), respectively. Despite the overall falling trend, the annual and seasonal mean C-factor value varied from year to year and three sub-periods could be distinguished: the period 1969–1993 when the annual mean C-factor value descended from a relatively high level to below average, the period 1994–2003 characterized by a sharp steady increase ($p < 0.05$), and the recent period 2004–2019 during which it decreased to its minimum ($p < 0.05$). In winter, the variation in C-factor value was characterized by a large fluctuation range and nonsignificant downward trend, with a downward rate of 0.02/a. In terms of seasonal scale, the multi-year average C-factor values were 17.53, 16.37, 7.52, and −3.12, respectively, indicating that the seasonal performance of wind erosion climatic erosivity was spring > summer > autumn > winter, so spring and summer were the high-risk period of soil wind erosion in this area.

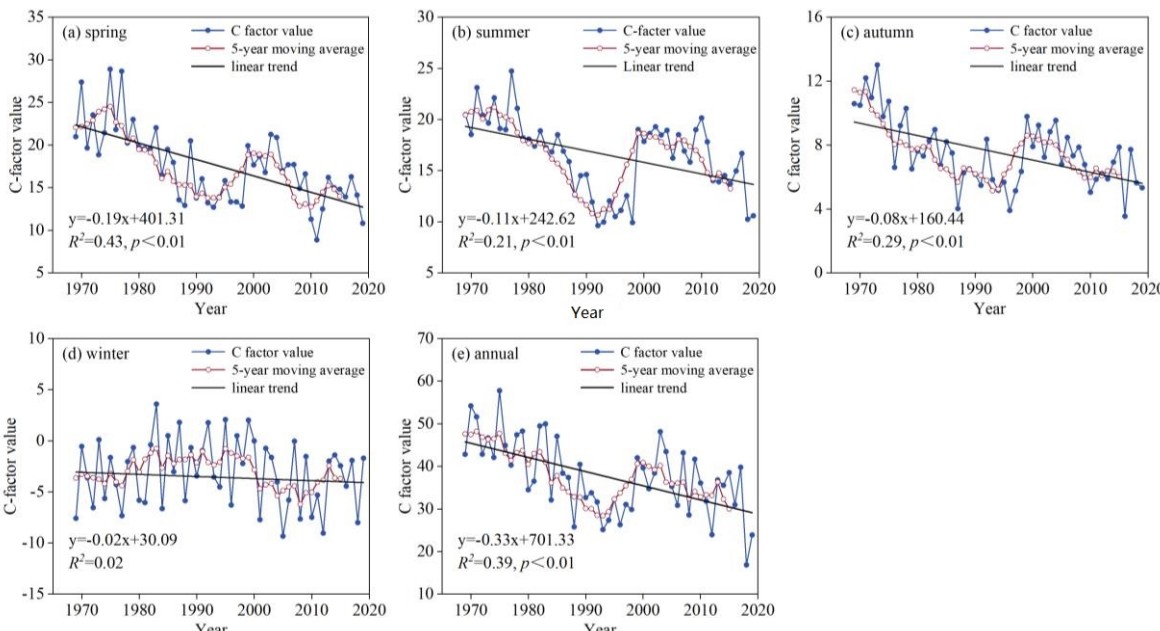

**Figure 2.** Variance tendency of C-factor value in (**a**–**d**) seasonal and (**e**) annual scales (50 years).

Table 1 presents C-factor value statistics of the stations with significant and nonsignificant trends at the 95% confidence level based on the significance test for the 64 stations during 1969–2019. At 36 stations or in 56% of Xinjiang, the annual C-factor value showed a significant downward trend at the 0.05 significance level, and at 14 stations or in 21% of the region, a nonsignificant decline trend. At 24 stations, mainly distributed in the Middle

East of Xinjiang, it displayed a significantly (4 stations) and nonsignificantly (10 stations) increasing trend. Statistics of the variation trend showed that the spring, summer, autumn, and winter climatic erosivity all tended to decline overall for most of the weather stations. From spring to winter, the number of stations with a noticeable downward trend of C-factor value decreased by 25%, and the number of stations with a prominent upward tendency increased by 17%.

**Table 1.** C-factor value statistics of the stations with variation trend of climatic erosivity (50 years).

| Time | Upward Trends | Nonsignificant Increase | Significant Increase | Downward Trends | Nonsignificant Decrease | Significant Decrease |
|---|---|---|---|---|---|---|
| Spring | 10 | 8 | 2 | 52 | 10 | 42 |
| Summer | 10 | 6 | 4 | 52 | 13 | 39 |
| Autumn | 16 | 10 | 6 | 46 | 11 | 35 |
| Winter | 24 | 11 | 13 | 38 | 22 | 16 |
| Annual | 14 | 10 | 4 | 50 | 14 | 36 |

*3.2. Spatial Distribution of Annual Climatic Erosivity*

Due to the spatial heterogeneity of the relevant climate variables, the mean annual C-factor varied greatly in space (Figure 3). The C-factor value during 1969–2019 showed an obvious spatial variation. The distribution of annual climate erosion was roughly bounded by 85° E. At stations mainly distributed in the west of the boundary, which covered 42.2% of the region, climatic erosivity was very weak with the C-factor value generally less than 30. At 67 stations, which were concentrated in eastern Xinjiang and represented 25.4% of the region, climatic erosivity was strong with the C-factor value exceeding 50. Regions with high values ($C \geq 100$) were concentrated in the junction between the Turpan and Hami Basin, which is situated on the east part of the study area. The spatial distribution of climate inclination rates exhibited an obvious north–south difference. In Northern Xinjiang and areas along Tianshan Mountain, the tendency rate was positive, and the C-factor value showed an upward trend. In southern Xinjiang and the central part of northern Xinjiang, the tendency rate was negative, and the C-factor value showed a downward trend, especially in the southwest edge of the Jungar basin.

On a seasonal scale, the spatial distribution characteristics of C-factor value in spring, summer, and autumn were similar, and the distribution characteristics of climatic erosivity roughly showed a zonal distribution along the longitude direction. In winter, the climatic erosivity presented a layered distribution along the latitude direction. Comparing the distribution of C-factor value and its tendency rate in each season, the severe erosion area was most widely distributed in spring and significantly narrowed in summer and autumn, and the erosivity was weakest in winter. The eastern and southwestern regions of Xinjiang and the Junggar Basin were areas with strong wind erosion climate erosion activity in spring and summer, while the Tuha basin, Taklimakan Desert, and its surrounding areas were the main areas for soil wind erosion control in winter.

In spring, summer, and autumn, at more than 56% of the stations, which were distributed in Altay and Tacheng, the climatic erosivity showed a significant downward trend. At stations mainly distributed in the west and south of the Taklimakan Desert, which covered 16–25% of the region, climatic erosivity was on the rise. Regions with an upward tendency of C-factor value were concentrated in Northern Xinjiang, which covers 39% of the region. The decline rate of climatic erosivity in Taklimakan Desert and eastern Xinjiang was the most obvious, accounting for 26% of the total stations. It was found that higher climate tendency rates often corresponded to higher climatic erosivities in autumn and winter; and in spring and summer, higher climate tendency rates corresponded to lower climatic erosivities.

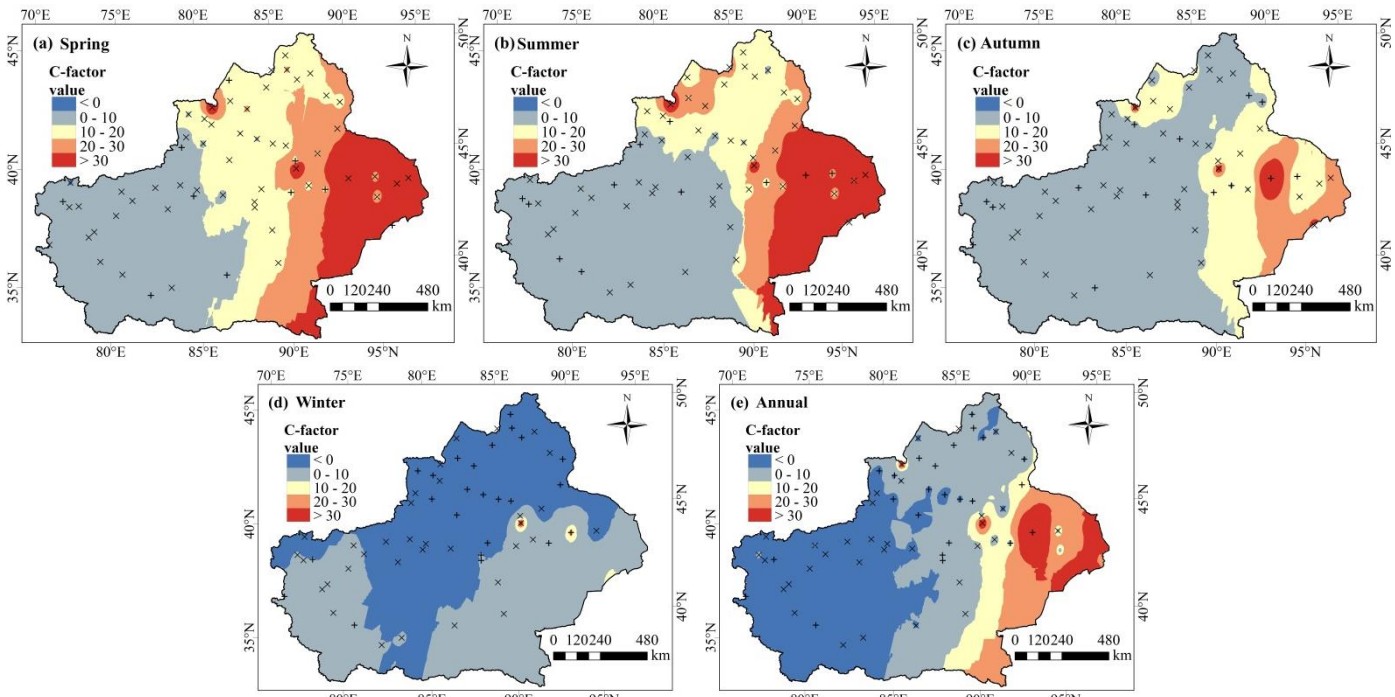

**Figure 3.** Spatial distribution of C-factor value on (**a**–**d**) seasonal and (**e**) annual scales. '+' and '×' denote upward and downward trends, respectively (50 years).

### 3.3. Climatic Erosivity Anomalies Associated with ENSO

Table 2 shows data on wind erosion climatic erosivity in Xinjiang during El Niño and La Niña events, revealing that the average climatic erosivity during El Niño years is lower than during La Niña years. According to statistical analysis, the average monthly C-factor value was 3.83. During the El Niño/La Niña period, the C-factor value was 3.21, which was somewhat lower than the average C-factor value from 1969 to 2016. In cold events, the C-factor value reached a low of 1.94 in 1995, a high of 4.75 in 1973, and an average of 3.42. The C-factor value reached a low of 1.60, a maximum of 5.03, and an average of 3.07 during the warm events. In conclusion, climatic erosivity during the neutral period was higher than during the ENSO period. The climatic erosivity during the warm events was 64% of that during the neutral era.

**Table 2.** Average monthly climatic erosivity during El Niño and La Niña events (50 years).

| Events | Time Span | Average C-Factor Value | Events | Time Span | Average C-Factor Value |
|---|---|---|---|---|---|
| El Niño | 1969.01–1970.01 | 4.60 | La Niña | 1995.08–1996.03 | 1.94 |
| La Niña | 1970.07–1972.01 | 4.62 | El Niño | 1997.05–1998.05 | 3.19 |
| El Niño | 1972.05–1973.03 | 4.60 | La Niña | 1998.07–2001.03 | 3.86 |
| La Niña | 1973.06–1976.03 | 4.75 | El Niño | 2002.06–2003.02 | 3.48 |
| El Niño | 1976.09–1977.02 | 2.26 | El Niño | 2004.07–2005.04 | 3.51 |
| El Niño | 1977.09–1978.01 | 2.64 | El Niño | 2006.09–2007.01 | 2.05 |
| El Niño | 1979.10–1980.02 | 1.90 | La Niña | 2007.08–2008.06 | 3.30 |
| El Niño | 1982.04–1983.06 | 5.03 | El Niño | 2009.07–2010.04 | 3.27 |
| La Niña | 1984.10–1985.06 | 3.95 | La Niña | 2010.07–2011.04 | 3.31 |
| El Niño | 1986.09–1988.02 | 3.07 | La Niña | 2011.08–2012.02 | 2.20 |
| La Niña | 1988.05–1989.05 | 3.81 | La Niña | 2014.11–2016.03 | 2.90 |
| El Niño | 1991.06–1992.07 | 3.03 | La Niña | 2017.09–2018.02 | 2.35 |
| El Niño | 1994.10–1995.03 | 1.60 | El Niño | 2018.10–2019.05 | 2.01 |

The ENSO does not occur in the whole time series. There was some bias in the correlation analysis between the full-time series SST index and climatic erosivity. As a result, the year of ENSO was extracted as time series in this study and examined with the climatic erosivity in the corresponding eras. Table 3 shows the results of the correlation analysis between the C-factor value of the same period and 1–7 months lagged and the SST monthly sequence value. The results of the analysis showed that the effect of El Niño and La Niña events on the wind erosion climatic erosivity in Xinjiang were different in duration.

**Table 3.** Correlation between climatic erosivity and SST in ENSO years.

| Lag Phase | El Niño | | La Niña | |
|---|---|---|---|---|
| | Correlativity | Significance | Correlativity | Significance |
| Corresponding period | −0.31 ** | 0.00 | 0.12 | 0.13 |
| One-month lag | −0.34 ** | 0.00 | 0.11 | 0.18 |
| Two-month lag | −0.30 ** | 0.00 | 0.05 | 0.51 |
| Three-month lag | −0.15 | 0.06 | −0.09 | 0.27 |
| Four-month lag | −0.02 | 0.79 | −0.18 * | 0.03 |
| Five-month lag | 0.10 | 0.20 | −0.31 ** | 0.00 |
| Six-month lag | 0.15 | 0.06 | −0.32 ** | 0.00 |
| Seven-month lag | 0.14 | 0.08 | −0.23 ** | 0.00 |

Note: ** indicates significance at $p < 0.01$; * indicates significance at $p < 0.05$; n.s.—nonsignificant.

In the El Niño event period and the 1–4 months after its end, there was a negative correlation between the SST and C-factor value in Xinjiang, and the correlation was the most obvious in the lag of 1–2 months. Five months after the end of the El Niño events, SST showed a positive but insignificant correlation with climatic erosivity. In the year of La Niña events, the correlation between SST and climatic erosivity showed the following characteristics: in the occurrence stage of La Niña events and 1–3 months after its end, they exhibited a positive but insignificant correlation. There was a very significant negative correlation in the 5–7 months behind.

To summarize, the influence of ENSO occurrences on the climate in Xinjiang did not dissipate soon after the event ended. The influence of El Niño events on climatic erosivity in Xinjiang continued from the commencement of the events until two months after its end. Our study also discovered an evident lag relationship between La Niña events and wind erosion climatic erosivity; specifically, the influence of La Niña events on climatic erosivity began to manifest in the fourth month after its end and began to fade after four months.

This study further explored the relationship between climate indices (ENSO, NAO, and AO) and climatic erosivity in Xinjiang utilizing the cross-wavelet transform (XWT) and wavelet coherence transform (WTC).

The XWT between ENSO and C-factor value (Figure 4a) showed a common power in the band of 3–4 years from 1983 to 1987 and the 4–5 years band in 1998–2002, whereas it was mainly anti-phase in the band of 4–5 years, and slightly shifted to the place where SST leads the C-factor value in the period of 3–4 years. Compared with XWT, there was a larger area of WTC of ENSO and C-factor value (Figure 4b) as being significant against the red background noise with a 5% significant level. Between 1995 and 2008, the ENSO and the C-factor value displayed strongly coherent periods of 8–12 years, with the phase difference vector tilted upward, indicating that, on that timescale, the ENSO led the C-factor.

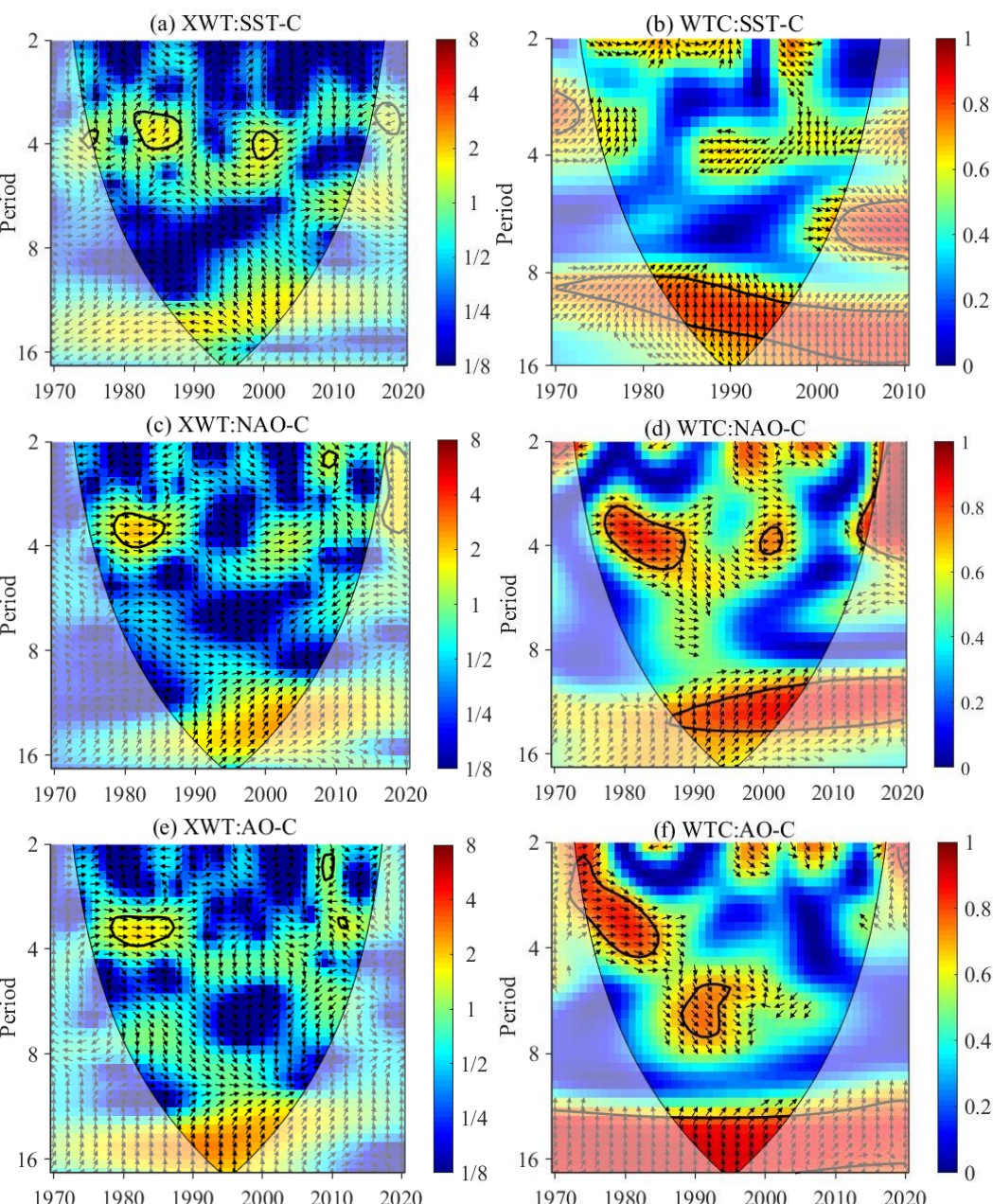

**Figure 4.** The cross-wavelet power (**a**) and the wavelet coherency (**b**) between the Niño-Southern Oscillation (ENSO) and C-factor value; the cross-wavelet power (**c**) and the wavelet coherency (**d**) between the North Atlantic Oscillation (NAO) and C-factor value; the cross-wavelet power (**e**) and the wavelet coherency (**f**) between the Arctic Oscillation (AO) and C-factor value. Contours are for variance units. The vectors indicate the phase difference between the multivariate climatic index and C-factor value. In all panels, the thick black line is the 5% significance level using the red noise model, and the thin black line indicates the cone of influence.

As shown in Figure 4c, there was one band with a good correlation between the NAO and C-factor value, which revealed a 3–4 year period from 1980 to 1986. The arrows pointed to the right (positive phases), suggesting the positive correlation between NAO and climatic erosivity. The WTC of NAO and the C-factor value showed a large area being significant in the bands of 3–5 from 1978 to 1990 and the 1–15 period from 1985 to 2005 (Figure 4d), and the arrows pointed to the right and northeastward, which revealed a consistent in-phase relationship in the significant bands between the two variables.

Figure 4e presented a consistent in-phase relationship in the period of 3–4 years from 1977 to 1985, implying a positive correlation between AO and climatic erosivity. There were two significant bands in the WTC between the AO and C-factor value (Figure 4f), which indicated a 2.2–7.8 year period from 1972 to 1995 and a 12–16 year period from 1985 to 2005. Most of the arrows pointed to the east, implying that the AO and climatic erosivity was in phase.

## 4. Discussion

There are seasonal changes in climatic erosivity in distinct places due to the effect of the monsoon climate. According to this study, the wind erosion climatic erosivity in Xinjiang was highest in the spring and summer, gradually dropped in the autumn, and reaches its lowest in the winter. The emergence of this result was inseparable from the characteristics of climate in Xinjiang, and is more consistent with previous research results (the maximum value of wind erosion climate factor index appeared in spring) [40,41]. In spring, the temperature in Xinjiang began to warm up, and the soil began to thaw, resulting in a loose soil structure. In addition, during this period, plants grew slowly and were in a state of wilting, which weakened the protection of vegetation on the surface, and the surface was exposed or semi-exposed, thus creating extremely favorable conditions for wind erosion on the surface. The minimum values of climatic erosion in arid and semi-arid regions of China, the Yarlung Zangbo River Basin, and Alashan Plateau appear in summer and autumn [12,42], while in Xinjiang, the minimum value occurs in winter. The reason was that the average temperature in Xinjiang is about 20 °C below zero in winter, and it is not easy for snow to melt and cover the bare surface, so it is not easy to cause surface wind erosion [43]. Therefore, spring and summer is a critical period to implement soil conservation practices. On the seasonal scale, sandstorms occur frequently in winter and spring. However, the results of this paper showed that the climatic erosivity is weak in winter. The reason for the deviation of the two conclusions may be that soil wind erosion is the result of the interaction of many influencing factors. In addition to the climatic factors proposed in this paper, it is also related to the physical and chemical properties of soil, roughness, vegetation coverage, and so on.

The reduction in wind speed and gale days, and climatic warming and humidification directly resulted in the decrease in wind erosion climatic erosivity [44]. During the past 50 years, the weakening of westerly circulation and winter monsoon and the decrease in the intensity and frequency of cold air activity resulted in the surface wind speed on the wane in the north of China [45,46]. In addition, the decrease in pressure gradient and weakness of the continental cold high pressure were also important reasons for the decrease in wind speed. The study of Zhao et al. also confirmed that the weakening of polar vortex intensity and area index in the northern hemisphere in the recent 49 years gave rise to the weakening of cold air activity intensity and frequency in Xinjiang [47], which led to the significant reduction in gale days in Xinjiang.

In various seasons, the distribution of climatic erosivity changed substantially. In spring, the distribution range of the severe erosion area was broadest, and the distribution of the high value area of climatic erosivity in this study was in accordance with that reported by reference [48]. Small differences were mainly because of the number of or the subtle difference in the time periods for the meteorological stations. According to the findings of this study, the center and western parts of northern Xinjiang and eastern Xinjiang had a high C-factor value, whereas the climatic erosivity in southern Xinjiang was modest. The interaction between atmospheric conditions and complex local terrain may affect the spatial distribution of the C-factor value to some extent. The northern and southern mountains in the region hinder the transportation of water vapor, and the central region is blocked by the Tianshan Mountains, which makes the climate conditions in the northern and southern regions of Xinjiang vary greatly. However, related research has indicated that sand-dust storms occurred frequently in southern Xinjiang and soil wind erosion is the primary link of sandstorms [49]. The main reason for the above deviation

was that the spatial distribution of wind erosion does not necessarily depend entirely on climatic conditions, and the distribution of the sand source on the underlying surface is another important factor affecting wind erosion.

Climatic erosivity varied due to a variety of climatic parameters such as wind speed, precipitation, temperature, and drought. However, recent evidence has shown that the temperature and precipitation in Northwest China were correlated with tropical SST anomalies [50,51]. Extensive studies have pointed out that the annual and monthly drought indices in Northwest China were closely connected to AO and ENSO events, respectively [52]. This study focused on the analysis of the variation characteristics of climatic erosivity in Xinjiang during ENSO events and found that climatic erosivity in the La Niña period was greater than that in the El Niño period. Previous studies have shown that the occurrence of El Niño events had a humidifying effect in Xinjiang [28]. On the contrary, during the La Niña event, the precipitation in Xinjiang showed a gradually decreasing trend, and the temperature was mainly high in the seven months after the end of the cold events [53,54]. Therefore, after the end of the La Niña events, the drought trend in Xinjiang increased, so the soil wind erosion was relatively large during the La Niña events.

In our study, correlations between the climatic erosivity and the three climate indices (ENSO, NAO, and AO) were analyzed. From a statistical perspective, the three climate indices showed relationships to the climatic erosivity in Xinjiang in terms of their correlation and periodicity. This was mainly because Xinjiang is located in the westerly-dominated climatic regime, and climatic conditions were impacted by the latitude wave propagation of the mid-latitude atmospheric circulation [55]. Correlations between the climatic erosivity and the three climate indices were neither simple linear relationships nor simple positive or negative relationships; rather, they were related to an advance or delay in the cycle. This also indicates the complexity of the factors that affect climatic erosivity in Xinjiang, though the exact mechanisms involved in this require further study. The limitation of this study is that the selected climatic index is limited, which cannot fully reveal the influence of the atmospheric circulation model on the wind erosion climatic erosivity in Xinjiang. In addition, there were many indicators that can characterize the ENSO phenomenon, including the MEI index, SOI index, and SST index. In this paper, only the SST index was used to characterize the ENSO phenomenon. Therefore, the research conclusion needs further in-depth research to verify.

## 5. Conclusions

In the past 50 years, the wind erosion climatic erosivity in Xinjiang had fluctuated and decreased at annual and seasonal scales. The C-factor value in spring, summer, and autumn exhibited a predominant downward tendency, but the C factor value in winter displayed a stable condition with an indistinctive downward trend. Other than winter, climatic erosivity exhibited a comeback tendency throughout the 1990s. The intensity of climatic erosivity was highest in the spring and summer and reached its lowest in the winter. Spring and summer were the high-risk periods of soil wind erosion in Xinjiang.

At the annual scale, climatic erosivity was significantly higher in the Eastern Junggar Basin and Junction between the Turpan Basin and Hami Basin, while it was lower in the Altay Prefecture and the desert area in southern Xinjiang. Eastern Xinjiang, the Junggar Basin, and its southwestern margin were areas with strong climatic erosion activities in spring and summer, while in winter, Taklimakan Desert and its surrounding areas were the main regions for prevention and control of soil wind erosion activities. The tendency rates of the C-factor value in spring, summer, autumn, and winter were −1.8/10a, −0.7/10a, −0.6/10a, and −0.1/10a, respectively. In autumn and winter, the climatic tendency rate was high in the regions with a high intensity of climatic erosivity. In spring and summer, the higher climate tendency rate corresponded to a lower climatic erosivity.

The average C-factor value was weaker during El Niño events and stronger during La Niña events, which implies that the climatic erosivity was controlled by large-scale atmospheric circulations. The impact of ENSO events on the Xinjiang climate did not

disappear immediately with the end of events. The influence of El Niño events on climatic erosivity in Xinjiang could be sustained from the beginning of the events to two months after the end of the events. La Niña events had a lag effect on the climatic erosivity in Xinjiang, with a lag period of 4 months. However, not all ENSO events were responsible for climatic erosivity during this period, and other drivers remain to be identified.

From a statistical perspective, the climate indices (ENSO, NAO, and AO) were related to climatic erosivity in Xinjiang in terms of correlation and periodicity. However, correlations between the climatic erosivity and climate indices were neither simple linear relationships nor definitely positive or negative relationships; rather, they were related to an advance or delay in the phase. From the cross-wavelet transform of data pairs, it was found that the ENSO led the climatic erosivity. The NAO and climatic erosivity revealed a consistent in-phase relationship in the significant bands. The AO and climatic erosivity was in phase.

**Author Contributions:** Y.W. worked on data collection, data analysis, and writing—original draft; H.Y. and W.F. conceived this study; C.Q. and K.S. helped in the result interpretation and write-up. All authors have read and agreed to the published version of the manuscript.

**Funding:** This work was supported by the National Natural Science Foundation of China (51569028, 41761064 and 51769030) and the International S&T Cooperation Program of China (2011DFA32800).

**Institutional Review Board Statement:** Not applicable.

**Informed Consent Statement:** Not applicable.

**Data Availability Statement:** Not applicable to this article, as there are no data that were generated.

**Acknowledgments:** The authors thank the National Meteorological Information Centre of China and National Geomatics Center of China for providing past climate data and base map data, respectively, of the study districts free of charge. We thank the anonymous reviewers and the editors of the journal for their constructive comments, suggestions, and edits to the manuscript.

**Conflicts of Interest:** The authors declare no conflict of interest.

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
