# Peer review of "Dynamic Variability of Wind Erosion Climatic Erosivity and Their Relationships with Large-Scale Atmospheric Circulation in Xinjiang, China"

_atmosphere, doi:10.3390/atmos13030419_

Round 1

Reviewer 1 Report

An interesting study that quantifies annual and seasonal wind erosion in Xinjiang, China using C-factor.

My questions and recommendations are shown below in accordance to page numbers:

page 1, line 17: add (50 years) after 1969-2019

page 1, line 41 [9,10] also I would recommend adding this recently published article that relates to the effect of climatic conditions on wind erosion and dust emission:

Li J., Garshick E., Al-Hemoud A., Huang S., Koutrakis, P. Impacts of meteorology and vegetation on surface dust concentrations in Middle Eastern countries. Science of Total Environment. 2020; 712: 136597 

page 2, line 62 [21,22] also I would recommend adding this recently published article that relates to global El-Nino:

Li, J., Garshick, E., Huang, S., Koutrakis, P., 2021. Impacts of El Niño-Southern Oscillation on surface dust levels across the world during 1982–2019. Science of The Total Environment 769, 144566

page 2, line 82 and page 3 line 116: add (50 years) after 1969-2019

page 3: lines 103-104: the phrase: "and the environment has a weak capacity to delay and contain disasters" is not understood, please re-write.

In the titles of Figures 2, 3, 4,  and Tables 1 and 2: Please add (50 years) in the end of the title.

page 5, line 202: should be written as: Table 1 presents C-factor value statistics.....

Page 6, Table 1, line 213: the title should be written as: C-factor value statistics.....

page 7, Table 2, line 265: The title of the table should clarify El Nino and La Nina 

Reviewer 2 Report

This is an interesting study. Nevertheless, it needs some further improvements. In general, there are still some occasional grammar errors throughout the manuscript, especially the article "the," "a," and "an" is missing in many places; please make a spellchecking in addition to these minor issues. The reviewer has listed some specific comments that might help the authors further enhance the manuscript's quality.

  1. Specific Comments
  • A list of acronyms is needed

Introduction

  • The objectives should be more explicitly stated.
  • What is the novelty of this work?

  • Methods
  • The methodology limitation should be mentioned.
  • All variables should be explained.

  • Results
  • This section is well written.

  • Discussion
  • Overall, the discussion part is week. The Discussion should summarize the manuscript's main finding(s) in the context of the broader scientific literature and address any limitations of the study or results that conflict with other published work.
